# Discovery of 2-(1-(3-(4-Chloroxyphenyl)-3-oxo- propyl)pyrrolidine-3-yl)-1*H*-benzo[d]imidazole-4-carboxamide: A Potent Poly(ADP-ribose) Polymerase (PARP) Inhibitor for Treatment of Cancer

**DOI:** 10.3390/molecules24101901

**Published:** 2019-05-17

**Authors:** Rui Min, Weibin Wu, Mingzhong Wang, Lin Tang, Dawei Chen, Huan Zhao, Cunlong Zhang, Yuyang Jiang

**Affiliations:** 1Department of Chemistry, Tsinghua University, Beijing 100084, China; mr16@mails.tsinghua.edu.cn; 2The Ministry-Province Jointly Constructed Base for State Key Lab-Shenzhen Key Laboratory of Chemical Biology, The Graduate School at Shenzhen, Tsinghua University, Shenzhen 518055, China; 3Shenzhen Kivita Innovative Drug Discovery Institute, Shenzhen 518055, China; weibin.wu@szkivita.com (W.W.); mzwang2000@163.com (M.W.); adamtl@126.com (L.T.) dawei-ba@163.com (D.C.); 4Shenzhen Ruikang Pharmaceutical Technology Co. Ltd., Shenzhen 518055, China; zhaohuan8650@126.com; 5Department of Pharmacology and Pharmaceutical Sciences, School of Medicine, Tsinghua University, Beijing 100084, China

**Keywords:** poly(ADP-ribose) polymerase, PARP enzyme inhibition, benzimidazole carboxamide

## Abstract

A series of benzimidazole carboxamide derivatives have been synthesized and characterized by ^1^H-NMR, ^13^C-NMR and HRMS. PARP inhibition assays and cellular proliferation assays have also been carried out. Compounds **5cj** and **5cp** exhibited potential anticancer activities with IC_50_ values of about 4 nM against both PARP-1 and PARP-2, similar to the reference drug veliparib. The two compounds also displayed slightly better *in vitro* cytotoxicities against MDA-MB-436 and CAPAN-1 cell lines than veliparib and olaparib, with values of 17.4 µM and 11.4 µM, 19.8 µM and 15.5 µM, respectively. The structure-activity relationship based on molecular docking was discussed as well.

## 1. Introduction

Poly(ADP-ribose) polymerase-1 (PARP-1) is a kind of enzyme closely involved in the DNA damage repair process. The PARP family contains 18 subtypes, while only PARP-1 and PARP-2 contain a DNA binding domain which facilitates the recognition and localization of the DNA damage site [1]. In DNA damage repairing process, PARP-1 and PARP-2 catalyze the degradation of nicotinamide adenine dinucleotide (NAD^+^) to nicotinamide and ADP-ribose, and the synthesis of poly(ADP-ribose) on the acceptor proteins by the formed ADP-ribose as substrate, which is the essential process in the repair of DNA [2,3,4]. The inhibition of PARP will cause synthetic lethal effects in cells, which makes PARP a hot target in cancer therapy [5].

Because of the similarities in structure to the natural substrate nicotinamide, such heterocyclic derivatives as quinazoline, phenanthridone, phthalazine and benzimidazole were developed as different generations of PARP inhibitor scaffolds [4,6,7]. so far, there are four approved parp inhibitors—olaparib, Niraparib, Rucaparib and Talazoparib (Figure 1)—which are on the market as chemotherapy drugs. Olaparib, based on the 2*H*-phthalazin-1-one scaffold, was the first FDA-approved oral PARP inhibitor drug for therapy of BRCA-mutated cancer in women with recurrent ovarian cancer [8,9]. The clinical trial of the drug combination of olaparib and taxol for terminal gastric cancer treatment finally failed in 2018. Niraparib, a PARP inhibitor derived from the indazole carboxamide lead developed by Merck, was approved by the FDA in 2017 for treatment of Pt-sensitive recurring (PSR) ovarian and peritoneal cancers without the limitation of BRCA mutations [10,11,12]. Besides competitively inhibiting catalytic active site of PARP-1, niraparib has the strongest ability to capture PARP-1 on DNA chains [13]. Rucaparib, another PARP inhibitor on the market derived from tricyclic indolactam, was the first PARP inhibitor put into clinical trials, and then approved in 2016 by the FDA for the treatment of BRCA-mutated terminal ovarian cancer [14]. Rucaparib was approved for maintenance treatment of PSR ovarian cancer in 2018. Talazoparib, consisting of a 2*H*-phthalazin-1-one scaffold, like olaparib, was approved for the treatment of patients carrying germ line BRCA-mutated, HER2 negative ovarian cancer in late 2018 [15].

The benzimidazole carboxamide scaffold is the basis of an effective series of PARP-1 inhibitors due to its relatively low molecular weight and high intrinsic potency, which made veliparib (ABT-888) a promising PARP inhibitor lead compound for antitumor drugs [14,16,17,18,19]. The amide group acts as an analogue of nicotinamide to bind with the PARP-1 active site. The intermolecular hydrogen bond interactions between the amide bond and Gly-863 and Ser-904 residues in the active site of PARP-1 contribute crucially to the potency, along with the π-π interaction with Tyr-907. However, as shown in the X-ray co-crystal structure of veliparib with PARP-1 (Figure 2), veliparib does not form distinct interactions with the residues in the hydrophobic pocket of PARP-1 [16]. In addition, veliparib exhibits a relatively low ability to trap PARP-1 on DNA and relatively low cellular activity in cancer cells, as compared with other PARP-1 inhibitors [13,20,21].

In this report, we designed a series of benzimidazole carboxamide derivatives based on veliparib, in which an aromatic ring substituting alkyl side chain was attached to the nitrogen atom in the five-member cyclic amine, in order to improve the combination of the compound and the active site of PARP. A phenyl group was attached to the terminal of the side chain for the sake of increasing the membrane permeability. Since PARP inhibitors will cause synthetic lethal effect specifically in BRCA-mutated cells, two cell lines, namely, MDA-MB-436 (a BRCA-1-mutated breast cancer cell line) and CAPAN-1 (a BRCA-2-mutated pancreatic cancer cell line) were selected to conduct cell proliferation assay. We discovered two potent PARP-1 and PARP-2 inhibitors exhibiting good potency in these two cell lines, against which veliparib and olaparib exhibited a relatively lower potency.

## 2. Results and Discussion

### 2.1. Chemistry

The benzimidazole ring was constructed by a ten-step large scale synthesis procedure as described previously (Scheme 1) [17,22]. Cbz-protected cyclic amine carboxylic ester **B1_6** was hydrolyzed to give acid **B1**. Then **B1** was coupled with 2,3-diaminobenzamide dihydrochloride under *N*,*N*′-carbonyldimidazole catalysis to give the amide product **N1**, which was then refluxed in acetic acid to produce Cbz-benzimidazole carboxamide **N2**. The Cbz protecting group was removed under hydrogenolysis conditions to provide a secondary amine **N3**. Nucleophilic substitutions using different chloro-substituted aromatic side chains on basic condition gave tertiary amine **N4**. Based on the structure of **N4**, we synthesized the target compounds.

### 2.2. PARP Inhibition Assay

In this report, we describe a series of benzimidazole carboxamide-containing PARP inhibitors in which a side chain has been introduced at the point of attachment of five-member cyclic amine expected to improve the activity. Table 1 shows the PARP-1 and PARP-2 inhibition assay results. The inhibition percentages at 10 nM of 16 compounds were measured, followed by IC_50_ values for the six compounds that showed relatively higher inhibition potency. Compounds **5cc**, **5ch**, **5ci**, **5cj**, **5co** and **5cp** showed relatively good PARP-1 inhibition potency, with IC_50_ values near or lower than 10 nM. By comparing IC_50_ of **5cd**, **5cp** and **5cc**, in which side chain contains 2-4 carbon length of phenylketone fragments*,*
**5cp** exhibited the much better potency than **5cc** and **5cd**, indicating that the style/position of carbonyl group, as well as its interactions with residues in the binding pocket may affect the inhibition. The docking study showed that **5cp** could bind to three important amino acid residues, Gly-863, Ser-904 and Glu-988, but **5cc** and **5cd** only interact with two of these residues, lacking the hydrogen bond with Glu-988. However, reduction of carbonyl group on **5cd** and **5cp** to a hydroxy group in **5cf** and **5cg** respectively, led to a dramatic decrease of the enzyme inhibition activities. The probable reason is that the binding pocket occupied by five-member ring prefers a lipophilic group (not shown in the docking diagram), so it is easy to understand why compounds **5cl** and **5cm** containing hydrophilic amine groups displayed lower enzyme potency. Moreover, **5ci, 5cj** and **5cp** with different substituents in the *para* position of the benzene ring exhibited much better enzyme potency that all the other compounds, as well as veliparib. Compounds **5ca** and **5cb, 5cn** and **5co** containing a *N*-phenylamine group or *N*-benzamide group, all displayed some enzyme inhibition activities, but slight lower than those phenylketone compounds **5cc, 5cd, 5ci, 5cj** and **5cp**. Compounds **5ce** and **5ck** both containing phthalimide group, a bulky group with large steric hindrance, displayed different enzyme inhibition activities, whose structure-activity relationship needs to be proved by more experiments.

### 2.3. Cell Proliferation Assay

Table 2 shows the results of cell proliferation assay of the analogues. Compared with compounds **5cd** and **5cc**, **5cp** showed the lowest IC_50_, which indicated that the side chain with three-carbon alkyl group exhibits the highest activity, while nitrogen atoms in the side chains decreased the potency, especially in CAPAN-1 cell line, as was shown in compounds **5ca**, **5cb** and **5co**. The terminal benzene group also played a vital part in exhibiting activity which facilitates membrane permeability, indicated in compounds **5cc**, **5cp**, **5cl** and **5cm**. Meanwhile, by comparing **5cc**, **5cp**, **5cf** and **5cg**, it was concluded that carbonyl group facilitated increasing the activity, since the reduction of the carbonyl to a hydroxy group greatly weakened the cellular potency. Compound **5cj** showed an even lower IC_50_ than **5cp**, which indicated that electron-withdrawing groups enhanced the activity, while electron-donating group in **5ci** impaired the activity, as compared with **5cp**.

### 2.4. Molecular Docking

The two compounds exhibiting the best potency, namely **5cj** and **5cp**, were chosen for molecular docking. As is shown in the molecular docking diagram of compound **5cj** with PARP-1 (Figure 3), the target molecule **5cj** was appropriately inserted into the catalytic active site of PARP-1. 

The amide group acts as hydrogen bond donor and acceptor to form intermolecular hydrogen bonds with Gly-863 and Ser-904, respectively. The electron-rich benzimidazole ring forms π-π stacking interactions with Tyr-907. The nitrogen atom in the *meta*-position of the amide group serves as a hydrogen bond donor to form a hydrogen bond with Glu-988. Compared with the X-ray co-crystal structure diagram of veliparib with PARP-1, the side chain of **5cj** was inserted into the hydrophobic pocket in the active site of PARP-1. Though the docking diagram of **5cj** did not show more intermolecular interactions distinctly besides the hydrogen bonding interactions existing in the crystal structure diagram of veliparib, there may exist Van der Waals interactions or hydrophobic interactions between the side chain and the residues in the active site of PARP-1, which facilitated the binding of **5cj** with PARP-1, and thus made **5cj** a better PARP-1 inhibitor, as compared with veliparib. Docking mode of **5cp** is similar to that of **5cj** (Figure 4), except the orientation of the side chain, which may be due to the influence of the chlorine atom in the benzene ring.

## 3. Materials and Methods

### 3.1. Genereral Informations

NMR spectra (^1^H 400 MHz, ^13^C 101 MHz) were obtained on Bruker 400 spectrometer (Karlsruhe, Germany) with the indicated solvent and internal standard. Chemical shifts are given in delta (δ) values and coupling constants (*J*) in Hertz (Hz). The following abbreviations are used for peak multiplicities: s, singlet; d, doublet; t, triplet; q, quartet; m, multiplet; br, broadened. Mass spectra were performed on a Waters Micromass Q-TOF Premier Mass Spectrometer (Milford, MA, USA) running as a flow injection acquisition. All solvents and reagents were obtained from commercial sources and used without further purification. Details for the ^1^H-NMR, ^13^C-NMR and HRMS of compounds **5ca**–**5cp** are provided in the section “Appendix A”.

### 3.2. Chemistry

#### 3.2.1. Procedure A: Synthesis of 2-(Pyrrolidin-3-yl)-1*H*-benzo[d]imidazole-4-carboxamide (**N3**)

*Step 1*: Preparation of *N*-benzyl-1-(trimethylsilyl)-methanamine (**B1_2**). A solution of B**1_1**(32 g, 261.1 mmol) and (chloromethyl)trimethylsilane (**C1**, 84 g, 783.9 mmol) in acetonitrile (500 mL) was stirred and heated at reflux overnight. The mixture was neutralized with 0.6 M NaOH solution (600 mL) and extracted by diethyl ether. The organic phase was dried over Na_2_SO_4_ and concentrated. The mixture was separated by column chromatography (silica gel, EtOAc/PE=1:10, Rf value 0.10) to give the title compound (20.62 g, 41%).

*Step 2*: Preparation of *N*-benzyl-1-methoxy-*N*-((trimethylsilyl)methyl)methanamine (**B1_3**). A solution of methanol (3.99 g, 124.1 mmol) and formalin (10.1 g, 124 mmol) was cooled to 0 °C. **B1_2** (20 g, 103.4 mmol) was added and the mixture was stirred for 0.5 h. The mixture was warmed to room temperature and stirred for 3 h. K_2_CO_3_ (14.3 g, 103.4 mmol) was added and the mixture was stirred for 1 h. The supernatant was transferred to another reaction flask. K_2_CO_3_ (5.0 g, 36.3 mmol) was added. The mixture was filtered, and the filter cake was rinsed by diethyl ether. The organic phase was concentrated to give the title compound (22.08 g, 89.9%).

*Step 3*: Preparation of methyl 1-benzylpyrrolidine-3-carboxylate (**B1_4**). A solution of **B1_3** (22.06 g, 92.7 mmol) and methyl acrylate (**C2**, 11.54 g, 139.0 mmol) in DCM (100 mL) was cooled to 0 °C. A solution of trifluoroacetic acid (12.71 g, 111.2 mmol) in DCM (50 mL) was added slowly by drop to the reaction mixture. The solution was warmed to room temperature and stirred for 17 h. NaHCO_3_ saturated solution (150 mL) was added to the solution and stirred until no gas was produced. The mixture was partitioned between water and DCM. The organic phase was washed with brine, dried over Na_2_SO_4_ and concentrated to give the title compound (19.53 g, 96%). ^1^H-NMR (CDCl_3_) δ 7.35–7.23 (m, 5H), 3.69 (s, 3H), 3.64 (s, 2H), 3.12–2.99 (m, 1H), 2.98–2.87 (m, 1H), 2.79–2.69 (m, 1H), 2.69–2.57 (m, 1H), 2.58–2.48 (m, 1H), 2.17–2.04 (m, 2H). MS (ESI, pos, ion): 219.9 [M + H]^+^.

*Step 4*: Preparation of methylpyrrolidine-3-carboxylate hydrochloride (**B1_5**). A solution of **B1_4** (19.50 g, 88.99 mmol), HCl dioxane solution (22.3 mL, 88.99 mmol) in methanol (150 mL) was treated with 10% Pd/C (1.95 g) and stirred at 50 °C under hydrogen atmosphere for 5 h. The solid was filtered off and the filtrate was concentrated to give the title compound (13.85 g, 94%).

*Step 5*: Preparation of 1-benzyl-3-methylpyrrolidine-1,3-dicarboxylate (**B1_6**). The solution of **B1_5** (13.85 g, 83.6 mmol) in NaHCO_3_ saturated aqueous solution (200 mL) was added by toluene (200 mL) and cooled to 0 °C. Benzyl chloroformate (13.83 mL, 83.6 mmol) was added by drop. The mixture was warmed to room temperature and stirred for 6 h, and then partitioned between water and toluene. The organic phase was washed with brine, dried over Na_2_SO_4_, and concentrated to give the title compound (20.03 g, 91%).

*Step 6*: Preparation of 1-((benzyloxy)carbonyl)pyrrolidine-3-carboxylic acid (**B1**). A solution of **B1_6** (20.03 g, 76.1 mmol) in THF (200 mL) was added by water (150 mL) and stirred. A solution of LiOH (3.64 g, 152.2 mmol) in water (100 mL) was added by drop. The mixture was heated and stirred at 60 °C for 5 h and the THF was removed. The mixture was extracted by EtOAc. And the aqueous phase was acidified to pH 2 by 2N HCl and partitioned between EtOAc and water. The organic phase was washed with brine, dried over Na_2_SO_4_ and concentrated to give the title compound (16.57 g, 87.4%).

*Step 7*: Preparation of benzyl 3-((2-amino-3-carbamoylphenyl)carbamoyl)pyrrolidine-1-carboxylate (**N1**). A solution of **B1** (16.57 g, 66.5 mmol) in pyridine (60 mL) and DMF (60 mL) was stirred with *N*,*N*′-carbonyldiimidazole (CDI, 11.76 g, 72.5 mmol) at 45 °C for 1 h. 2,3-Diaminobenzamide dihydrochloride (**A1**, 13.54 g, 60.4 mmol) was added and the mixture stirred at 45 °C temperature for 24 h. After concentration, the residue was partitioned between EtOAc and aqueous NaHCO_3_. The solid was collected and washed with water and dried to give the titled compound (11.97 g, 52%).

*Step 8*: Preparation of benzyl 3-(4-carbamoyl-1*H*-benzo[d]imi-dazol-2-yl)pyrolidine-1-carboxylate (**N2**). A suspension of **N1** (11.97 g, 31.3 mmol) in AcOH (85 mL) was heated at reflux for 2 h. After cooling, the solution was concentrated and the residue partitioned between EtOAc and aqueous NaHCO_3_. The organic layer was washed with water and concentrated to give the title compound (6.79 g, 59%).

*Step 9*: Preparation of 2-(pyrrolidin-3-yl)-1*H*-benzo[d]imidazole-4-carboxamide (**N3**). A solution of **N2** (6.79 g, 40 mmol) in MeOH (250 mL) was treated with 10% Pd/C (0.54 g) and stirred at 50 °C under hydrogen atmosphere for 5 h. The solid was filtered off and the filtrate was concentrated. The residue was stirred in petroleum ether and then filtered to give the titled compound (4.43 g, 100%). ^1^H NMR (400 MHz, DMSO) δ 7.52–7.30 (m, 2H), 7.13–6.94 (m, 1H), 3.77–3.33 (m, 2.5H), 3.24–3.01 (m, 2.5H), 2.39–2.21 (m, 1H), 2.11–1.96 (m, 1H). MS calcd for C_12_H_15_N_4_O, [M + H]^+^, 231.1, found 231.1.

#### 3.2.2. Procedure B: Synthesis of **5ca**, **5cb**, **5cc**, **5cd**, **5ce**, **5ch**, **5ci**, **5cj**, **5ck** and **5cp**

*2-(1-(2-Oxo-2-(phenylamino)ethyl)pyrrolidin-3-yl)-1H-benzo[d]imidazole-4-carboxamide* (**5ca**). A solution of **N3** (200 mg, 0.87 mmol), 2-chloro-*N*-phenylacetamide (221 mg, 1.30 mmol) in DMF (5 mL) was treated with K_2_CO_3_ (240 mg, 1.74 mmol) and KI (14.4 mg, 0.09 mmol) at 50 °C for 3 h. The solution was concentrated, and the residue was purified by column chromatography (silica gel, DCM/MeOH = 30:1, Rf value 0.22) to give the title compound (111 mg, 35%). ^1^H-NMR (methanol-*d*_4_) δ 7.86 (d, *J* = 7.6 Hz, 1H), 7.67 (d, *J* = 8.0 Hz, 1H), 7.54–7.49 (m, 2H), 7.31–7.24 (m, 3H), 7.11–7.05 (m, 1H), 3.85–3.76 (m, 1H), 3.52–3.39 (m, 2H), 3.17 (dd, *J* = 6.7, 2.4 Hz, 2H), 3.11–3.03 (m, 1H), 2.93–2.84 (m, 1H), 2.53–2.41 (m, 1H), 2.34 (m 1H). ^13^C-NMR (MeOD) δ 169.26, 158.72, 137.71, 128.44, 124.09, 122.23, 121.48, 119.98, 58.66, 58.29, 53.45, 37.37, 30.02. HRMS calcd for C_20_H_2__1_N_5_O_2_, [M + H]^+^, 364.1769, found 364.1768.

*2-(1-(3-Oxo-3-(Phenylamino)propyl)pyrrolidin-3-yl)-1H-benzo[d]imidazole-4-carboxamide* (**5cb**). The title compound was prepared according to procedure B using 4-chloro-1-phenylbutan-1-one in place of 2-chloro-*N*-phenylacetamide (21%). ^1^H-NMR (methanol-*d*_4_) δ 7.86 (d, *J* = 7.6 Hz, 1H), 7.63 (d, *J* = 8.0 Hz, 1H), 7.53–7.40 (m, 2H), 7.35–7.16 (m, 3H), 7.10–6.97 (m, 1H), 3.79 (dt, *J* = 13.1, 6.3 Hz, 1H), 3.20 (d, *J* = 7.4 Hz, 2H), 3.11–2.96 (m, 3H), 2.96–2.84 (m, 1H), 2.67 (t, *J* = 6.9 Hz, 2H), 2.53–2.28 (m, 2H). ^13^C-NMR (MeOD) δ 171.18, 169.20, 158.26, 138.29, 128.37, 123.79, 122.27, 121.48, 119.80, 58.22, 53.03, 51.21, 37.01, 34.86, 29.55. HRMS calcd for C_21_H_2__3_N_5_O_2_, [M + H]^+^, 378.1926, found 378.1926.

*2-(1-(4-Oxo-4-phenylbutyl)pyrrolidin-3-yl)-1H-benzo[d]imidazole-4-carboxamide* (**5cc**). The title compound was prepared according to procedure B using 4-chloro-1-phenylbutan-1-one in place of 2-chloro-*N*-phenylacetamide (40%). ^1^H NMR (methanol-*d*_4_) δ 8.07–7.97 (m, 2H), 7.88 (d, *J* = 7.7 Hz, 1H), 7.69 (d, *J* = 8.0 Hz, 1H), 7.65–7.57 (m, 1H), 7.50 (dd, *J* = 7.6 Hz, 2H), 7.32 (dd, *J* = 8.9, 6.7 Hz, 1H), 3.81–3.65 (m, 1H), 3.16 (dd, *J* = 11.1, 7.8 Hz, 2H), 3.04–2.77 (m, 4H), 2.74–2.61 (m, 2H), 2.50–2.37 (m, 1H), 2.35–2.23 (m, 1H), 2.02 (dd, *J* = 7.2 Hz, 2H). ^13^C-NMR (MeOD) δ 158.48, 132.79, 128.30, 127.72, 122.16, 121.39, 58.39, 55.02, 53.28, 48.44, 48.23, 36.97, 29.68, 22.85. HRMS calcd for C_22_H_24_N_4_O_2_, [M + H]^+^, 377.1973, found 377.1980.

*2-(1-(2-Oxo-2-phenylethyl)pyrrolidin-3-yl)-1H-benzo[d]imidazole-4-carboxa-mide* (**5cd**). The title compound was prepared according to procedure B using 2-chloro-1-phenylethan-1-one in place of 2-chloro-*N*-phenylacetamide in acetone solution (24%). ^1^H-NMR (DMSO-*d*_6_) δ 12.71 (br, 1H), 9.30 (br, 1H), 8.05–7.93 (m, 2H), 7.79 (d, *J* = 7.6 Hz, 1H), 7.73–7.59 (m, 3H), 7.51 (dd, *J* = 7.7 Hz, 2H), 7.25 (dd, *J* = 7.8 Hz, 1H), 4.21–4.03 (m, 2H), 3.79–3.64 (m, 1H), 3.26–3.13 (m, 1H), 3.02–2.85 (m, 2H), 2.79 (td, *J* = 8.6, 6.0 Hz, 1H), 2.37–2.15 (m, 2H). ^13^C-NMR (DMSO) δ 197.41, 167.05, 158.73, 136.19, 133.68, 133.25, 129.71, 129.11, 128.43, 122.39, 121.74, 114.79, 61.70, 58.92, 53.65, 40.43, 37.39, 29.47. HRMS calcd for C_20_H_20_N_4_O_2_, [M + H]^+^, 349.1660, found 349.1669.

*2-(1-(3-(1,3-Dioxoisoindolin-2-yl)propyl)pyrrolidin-3-yl)-1H-benzo[d]imidazole-4-carboxamide* (**5ce**). The title compound was prepared according to procedure B using 2-(3-bromopropyl)- isoindoline-1,3-dione in place of 2-chloro-*N*-phenylacetamide (57%). ^1^H-NMR (methanol-*d*_4_) δ 7.88–7.80 (m, 3H), 7.78–7.73 (m, 2H), 7.66 (dd, *J* = 8.0, 1.1 Hz, 1H), 7.28 (dd, *J* = 7.8 Hz, 1H), 3.84–3.76 (m, 2H), 3.70–3.60 (m, 1H), 3.11–3.02 (m, 1H), 2.98 (dd, *J* = 9.5, 6.5 Hz, 1H), 2.84–2.72 (m, 2H), 2.70–2.60 (m, 2H), 2.40–2.28 (m, 1H), 2.21–2.11 (m, 1H). ^13^C-NMR (MeOD) δ 169.26, 168.58, 158.51, 133.89, 132.00, 122.64, 122.19, 121.40, 58.28, 53.22, 53.03, 36.94, 35.83, 29.74, 26.83. HRMS calcd for C_23_H_2__3_N_5_O_3_, [M + H]^+^, 418.1875, found 418.1873.

*2-(1-(3-(Phenylamino)propyl)pyrrolidin-3-yl)-1H-benzo[d]imidazole-4-carboxamide* (**5ch**). The title compound was prepared according to procedure B using *N*-(3-chloropropyl)aniline in place of 2-chloro-*N*-phenylacetamide (28%). ^1^H NMR (400 MHz, DMSO-*d*_6_) δ 9.19 (br, 1H), 7.76 (d, *J* = 7.5 Hz, 1H), 7.62 (d, *J* = 7.5 Hz, 2H), 7.23 (t, *J* = 7.8 Hz, 1H), 7.00 (t, *J* = 7.7 Hz, 2H), 6.47 (dd, *J* = 16.1, 7.8 Hz, 3H), 5.54 (br, 1H), 3.66 (dd, *J* = 9.6, 6.9 Hz, 2H), 3.02 (dd, *J* = 7.0 Hz, 2H), 2.86–2.76 (m, 1H), 2.70 (dd, *J* = 7.9 Hz, 2H), 2.57 (dd, *J* = 7.4 Hz, 2H), 2.33–2.13 (m, 2H), 1.72 (p, *J* = 7.0 Hz, 2H).^13^C NMR (101 MHz, DMSO) δ 167.12, 159.00, 149.47, 129.29, 122.36, 121.72, 115.87, 112.41, 59.22, 55.32, 53.82, 41.70, 37.19, 29.92, 28.18. HRMS calcd for C_21_H_2__3_N_5_O_2_, [M + H]^+^, 364.2133, found 364.2138.

*2-(1-(3-(4-Methoxyphenyl)-3-oxopropyl)pyrrolidin-3-yl)-1H-benzo[d]imidazole-4-carboxamide* (**5ci**). The title compound was prepared according to procedure B using 3-chloro-1-(4-methoxy- phenyl)propan-1-one in place of 2-chloro-*N*-phenylacetamide (33%). ^1^H-NMR (chloroform-*d*) δ 11.36 (br, 1H), 9.65 (br, 1H), 8.14–7.93 (m, 3H), 7.64 (d, *J* = 7.9 Hz, 1H), 7.30 (dd, *J* = 4.9, 3.8 Hz, 1H), 7.04–6.89 (m, 2H), 3.90–3.83 (m, 3H), 3.80–3.70 (m, 1H), 3.36–3.07 (m, 6H), 2.84–2.71 (m, 1H), 2.55–2.35 (m, 2H), 2.12–2.01 (m, 1H). ^13^C-NMR (MeOD) δ 197.88, 197.83, 169.21, 163.98, 158.17, 130.20, 129.50, 122.24, 121.47, 113.51, 58.43, 54.67, 53.43, 50.53, 36.97, 36.37, 29.79. HRMS calcd for C_22_H_2__4_N_4_O_3_, [M + H]^+^, 393.1922, found 393.1919.

*2-(1-(3-(4-Chloroxyphenyl)-3-oxopropyl)pyrrolidin-3-yl)-1H-benzo[d]imidazole-4-carboxamide* (**5cj**). The title compound was prepared according to procedure B using 3-chloro-1-(4-chlorophenyl)- propan-1-one in place of 2-chloro-*N*-phenylacetamide (20%). ^1^H-NMR (chloroform-*d*) δ 11.37 (br, 1H), 9.63 (br, 1H), 8.03–7.84 (m, 3H), 7.63 (d, *J* = 7.9 Hz, 1H), 7.46–7.40 (m, 2H), 7.30–7.23 (m, 1H), 3.77–3.68 (m, 1H), 3.36–3.00 (m, 6H), 2.77 (dd, *J* = 9.4, 6.7 Hz, 1H), 2.54–2.35 (m, 2H), 2.11–1.99 (m, 1H). ^13^C-NMR (MeOD) δ 197.96, 169.22, 158.18, 139.22, 135.19, 129.44, 128.58, 122.24, 121.46, 120.12, 116.39, 58.45, 53.41, 50.09, 37.02, 30.55, 29.35. HRMS calcd for C_21_H_2__1_N_4_O_2_Cl, [M + H]^+^, 397.1427, found 397.1428.

*2-(1-(2-(1,3-Dioxoisoindolin-2-yl)ethyl)pyrrolidin-3-yl)-1H-benzo[d]imidazole-4-carboxamide* (**5ck**). The title compound was prepared according to procedure B using 2-(2-bromoethyl)- isoindoline-1,3-dione in place of 2-chloro-*N*-phenylacetamide (72%). ^1^H-NMR (methanol-*d*_4_) δ 7.88–7.78 (m, 3H), 7.78–7.72 (m, 2H), 7.62 (d, *J* = 8.0 Hz, 1H), 7.27 (dd, *J* = 7.8 Hz, 1H), 3.95–3.83 (m, 2H), 3.77–3.66 (m, 1H), 3.25–3.11 (m, 2H), 3.03–2.82 (m, 4H), 2.46–2.34 (m, 1H), 2.29–2.17 (m, 1H). ^13^C-NMR (MeOD) δ 169.19, 168.50, 158.24, 133.87, 132.03, 122.66, 122.21, 121.41, 58.20, 53.51, 53.23, 37.01, 36.13, 29.61. HRMS calcd for C_22_H_2__1_N_5_O_3_, [M + H]^+^, 404.1718, found 404.1719.

*2-(1-(3-Oxo-3-phenylpropyl)pyrrolidin-3-yl)-1H-benzo[d]imidazole-4-carboxamide* (**5cp**). The title compound was prepared according to procedure B using 3-chloro-1-phenylpropan-1-one in place of 2-chloro-*N*-phenylacetamide (77%). ^1^H-NMR (- DMSO-*d*_6_) δ 9.10 (br, 1H), 8.05–7.94 (m, 2H), 7.80 (d, *J* = 7.5 Hz, 1H), 7.77–7.60 (m, 3H), 7.60–7.48 (m, 2H), 7.34–7.19 (m, 1H), 4.08–3.96 (m, 1H), 3.93–3.79 (m, 1H), 3.72–3.60 (m, 3H), 3.60–3.43 (m, 4H), 2.62–2.49 (m, 1H), 2.44–2.27 (m, 1H). ^13^C-NMR (DMSO) δ 197.43, 166.96, 155.69, 136.48, 134.16, 129.38, 129.31, 129.03, 128.70, 128.53, 122.69, 122.36, 122.13, 56.90, 53.56, 49.87, 36.59, 35.11, 29.83. HRMS calcd for C_21_H_22_N_4_O_2_, [M + H]^+^, 363.1817, found 363.1815.

#### 3.2.3. Procedure C: Synthesis of **5cf** and **5cg**

*2-(1-(2-Hydroxy-2-phenylethyl)pyrrolidin-3-yl)-1H-benzo[d]imidazole-4-carboxamide* (**5cf**). A solution of **5cd** (30 mg, 0.14 mmol) in MeOH (5 mL) was treated by sodium borohydride (10 mg, 0.28 mmol) and stirred at room temperature for 4 h. The solution was concentrated, and the residue was extracted by EtOAc. The organic phase was concentrated to give the title compound (28 mg, 94%). ^1^H-NMR (methanol-*d*_4_) δ 7.87 (d, *J* = 7.6 Hz, 1H), 7.70–7.63 (m, 1H), 7.46–7.39 (m, 2H), 7.39–7.25 (m, 4H), 4.87–4.82 (m, 1H), 3.81–3.63 (m, 1H), 3.22–3.01 (m, 2.5H), 2.98–2.79 (m, 2.5H), 2.77–2.64 (m, 1H), 2.49–2.36 (m, 1H), 2.29–2.15 (m, 1H). ^13^C-NMR (MeOD) δ 169.29, 158.90, 143.36, 127.97, 127.16, 125.77, 122.17, 121.38, 100.00, 72.05, 63.33, 58.81, 53.76, 37.17, 29.35. HRMS calcd for C_20_H_2__2_N_4_O_2_, [M + H]^+^, 351.1817, found 351.1821.

*2-(1-(3-Hydroxy-3-phenylpropyl)pyrrolidin-3-yl)-1H-benzo[d]imi-dazole-4-carboxamide* (**5cg**). The title compound was prepared according to procedure C using **5cp** in place of **5cd** (72%). ^1^H-NMR (methanol-*d*_4_) δ 7.86 (dd, *J* = 7.7, 1.0 Hz, 1H), 7.67 (dd, *J* = 8.0, 1.1 Hz, 1H), 7.41–7.21 (m, 6H), 4.81–4.76 (m, 1H), 3.87–3.74 (m, 1H), 3.25–3.13 (m, 1H), 3.05–2.98 (m, 2H), 2.96–2.86 (m, 1H), 2.85–2.75 (m, 1H), 2.54–2.43 (m, 1H), 2.39–2.27 (m, 1H), 2.12–1.94 (m, 3H).^13^C-NMR (MeOD) δ 158.42, 144.70, 127.88, 126.84, 125.50, 122.18, 121.36, 72.80, 58.52, 53.36, 52.76, 37.03, 36.96, 29.28. HRMS calcd for C_21_H_2__4_N_4_O_2_, [M + H]^+^, 365.1973, found 365.1981.

#### 3.2.4. Procedure D: Synthesis of **5cl** and **5cm**

*2-(1-(3-Aminopropyl)pyrrolidin-3-yl)-1H-benzo[d]imidazole-4-carboxamide* (**5cl**).A solution of **5ce** (500 mg, 1.33 mmol) in EtOH (10 mL) was added by 80% hydrazine hydrate (832 mg, 13.3 mmol), and the solution was heated at reflux for 4 h. The solution was concentrated and the residue was purified by column chromatography (silica gel, MeOH/EtOAc = 1:1, Rf value 0.18) to give the title compound 199 mg (52%). ^1^H-NMR (methanol-*d*_4_) δ 7.86 (dd, *J* = 7.7, 1.1 Hz, 1H), 7.66 (dd, *J* = 8.0, 1.0 Hz, 1H), 7.28 (dd, *J* = 7.8 Hz, 1H), 3.80–3.65 (m, 1H), 3.11 (dd, *J* = 9.6, 8.0 Hz, 1H), 2.95 (dd, *J* = 9.6, 6.8 Hz, 1H), 2.88–2.71 (m, 4H), 2.71–2.53 (m, 2H), 2.46–2.35 (m, 1H), 2.33–2.18 (m, 1H), 1.75 (p, *J* = 7.3 Hz, 2H).^13^C-NMR (MeOD) δ 169.29, 158.58, 122.16, 121.40, 120.00, 116.49, 58.56, 53.48, 53.44, 39.49, 36.93, 30.50, 29.71. HRMS calcd for C_16_H_2__1_N_5_O, [M + H]^+^, 288.1820, found 288.1524.

*2-(1-(2-Aminoethyl)pyrrolidin-3-yl)-1H-benzo[d]imidazole-4-carboxamide* (**5cm**). The titled compound was prepared according to procedure D using **5ck** in place of **5ce** (210 mg, 62%). ^1^H-NMR (methanol-d_4_) δ 7.88–7.84 (m, 1H), 7.68–7.64 (m, 1H), 7.28 (dd, *J* = 7.8 Hz, 1H), 3.78–3.67 (m, 1H), 3.08 (dd, *J* = 9.5, 7.9 Hz, 1H), 2.97 (dd, *J* = 9.5, 6.5 Hz, 1H), 2.89–2.75 (m, 4H), 2.72–2.61 (m, 2H), 2.47–2.34 (m, 1H), 2.29–2.18 (m, 1H).^13^C-NMR (MeOD) δ 169.30, 158.75, 121.39, 119.94, 116.55, 58.68, 57.63, 53.36, 39.51, 37.01, 29.83. HRMS calcd for C_14_H_19_N_5_O, [M + H]^+^, 274.1664, found 274.1674.

#### 3.2.5. Procedure E: Synthesis of **5cn** and **5co**

*2-(1-(3-Benzamidopropyl)pyrrolidin-3-yl)-1H-benzo[d]imidazole-4-carboxamide* (**5cn**). A solution of **5cl** (150 mg, 0.52 mmol) in DMF (5 mL) was added by benzoyl chloride (95 mg, 0.68 mmol) in ice bath. The solution was stirred for 2 h, and then concentrated. The residue was purified by column chromatography (silica gel, DCM/EtOAc = 1:1, Rf value 0.24) to give the title compound (85 mg, 42%). ^1^H-NMR (methanol-*d*_4_) δ 7.87 (dd, *J* = 7.7, 1.0 Hz, 1H), 7.85–7.79 (m, 2H), 7.69 (dd, *J* = 8.0, 1.1 Hz, 1H), 7.56–7.49 (m, 1H), 7.47–7.41 (m, 2H), 7.31 (dd, *J* = 7.9 Hz, 1H), 3.99–3.88 (m, 1H), 3.60–3.48 (m, 4H), 3.30–3.17 (m, 2H), 3.11–3.01 (m, 2H), 2.65–2.52 (m, 1H), 2.44–2.32 (m, 1H), 2.09–1.97 (m, 2H). ^13^C-NMR (MeOD) δ 169.27, 156.57, 133.92, 131.42, 128.20, 126.87, 122.48, 121.75, 57.66, 53.56, 53.23, 37.09, 36.61, 29.66, 26.99. HRMS calcd for C_22_H_2__5_N_5_O_2_, [M + H]^+^, 392.2082, found 392.2085.

*2-(1-(2-Benzamidoethyl)pyrrolidin-3-yl)-1H-benzo[d]imidazole-4-carboxamide* (**5co**). The title compound was prepared according to procedure E using **5cm** in place of **5cl** (145 mg, 71%). ^1^H-NMR (methanol-*d*_4_) δ 7.93–7.85 (m, 3H), 7.68 (d, *J* = 7.9 Hz, 1H), 7.55 (d, *J* = 7.3 Hz, 1H), 7.47 (dd, *J* = 7.6 Hz, 2H), 7.31 (dd, *J* = 7.8 Hz, 1H), 4.12–4.03 (m, 1H), 4.01–3.87 (m, 2H), 3.80 (t, *J* = 5.9 Hz, 2H), 3.71–3.57 (m, 2H), 3.56–3.47 (m, 2H), 2.74–2.62 (m, 1H), 2.52–2.40 (m, 1H). ^13^C-NMR (MeOD) δ 169.64, 168.97, 155.15, 133.40, 131.73, 128.26, 127.12, 122.65, 121.95, 57.48, 55.25, 54.12, 36.51, 36.36, 29.49. HRMS calcd for C_21_H_2__3_N_5_O_2_, [M + H]^+^, 378.1926, found 378.1921.

### 3.3. PARP Inhibition Assay

The PARP-1 and PARP-2 inhibition assays were performed in outsourcing, by a CRO company, Shanghai Medicilon Inc. (Shanghai, China). The PARP-1 and PARP-2 inhibitory activities of the compounds were measured using PARP-1 Chemiluminescent Assay Kit (BPS Bioscience, catalog 80569, San Diego, CA, USA) and PARP-2 Chemiluminescent Assay Kit (BPS Bioscience, catalog 80552), respectively, according to the manufacturer’s instructions. Briefly, PARP-1 or PARP-2 biotinylated substrate was incubated with compounds or solvent control at varying concentrations and an assay buffer containing the PARP-1 or PARP-2 enzyme. After then, the plate was treated with streptavidin-HRP followed by addition of the HRP substrate and the luminescent signal was measured using a chemiluminescence reader (Perkin Elmer Envision 2104 Multi Label Microplate Reader, Waltham, MA, USA).

### 3.4. Cell Proliferation Assay

The MDA-MB-436 and CAPAN-1 cell lines were provided by Sundia MediTech Company, Ltd., Shanghai, China. A three-day assay was conducted. A cell suspension was prepared. Cell density was counted by an automated cell counter and then diluted to the required density, according to seeding density. 50 μL cells were seeded into 384-well plate in growth medium according to the plate map. The cells were incubated at 37 °C, 5% CO_2_ overnight.

200× compound solution in DMSO was prepared and diluted with growth medium to 26× final concentration by addition of 26 μL 200× compounds to 174 μL growth medium. Then 2 μL of diluted compound solution was added to cells and the cell was incubated at 37 °C, 5% CO_2_ for 72 h.

The assay plate was equilibrated to room temperature prior to measurement. 15 μL of CellTiter-Glo Reagent was added into each well and the contents were mixed for 2 min on an orbital shaker to induce cell lysis. The cells were incubated at room temperature for 60 min to stabilize luminescent signal. Luminescence was recorded on the Envision system.

### 3.5. Molecular Docking

The molecular docking experiment was based on the molecular modeling package SYBYL-X 1.3 (Tripos associate Inc., St. Louis, MO, USA). The 3D structure of PARP-1 receptor for the molecular docking study was downloaded from Protein Data Bank (PDB ID: 2RD6) [16]. The selected ligands were compound **5cj** and **5cp** in this experiment. The molecular docking experiment was implemented in four steps: (1) The water molecules in the protein crystal structure downloaded from PDB were deleted; (2) The energy of receptor and ligand was optimized; (3) The intrinsic ligand in the binding site of the receptor was deleted, and then the selected target compound was docked into the receptor according to Sybyl-X modules; (4) The molecular docking results obtained by the above steps were handled and analyzed.

## 4. Conclusions

In summary, a series of benzimidazole carboxamide derivatives have been synthesized and their PARP inhibition activities and cellular activities have been detected. These compounds displayed good PARP inhibition potency, as much as veliparib and olaparib. Compound **5cj** exhibited the best activity among 16 synthesized compounds, with IC_50_ of 3.9 nM against PARP-1 and 4.2 nM against PARP-2. Furthermore, compound **5cj** showed significant cytotoxicity against such two BRCA-mutated cell lines. To be specific, as for the cytotoxicity against MDA-MB-436, the IC_50_ of **5cj** is 17.4 μM, much lower than 30.2 μM of olaparib; and especially, as for the cytotoxicity against CAPAN-1, the IC_50_ of **5cj** is 11.4 μM, by contrast, veliparib and olaparib exhibited low potency within the three-day cell proliferation assay. Molecular docking study of **5cj** indicated that besides the hydrogen bond interactions formed by the benzimidazole carboxamide scaffold similar to Veliparib, the side chain of **5cj** being inserted into the hydrophobic pocket in the active site of PARP-1 leads to a different binding mode.

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
