# Peer review of "Discovery of 2-(1-(3-(4-Chloroxyphenyl)-3-oxo- propyl)pyrrolidine-3-yl)-1H-benzo[d]imidazole-4-carboxamide: A Potent Poly(ADP-ribose) Polymerase (PARP) Inhibitor for Treatment of Cancer"

_molecules, 2019, doi:10.3390/molecules24101901_

Round 1
Reviewer 1 Report
The paper entitled “2-(1-(3-(4-chloroxyphenyl)-3-oxopropyl)pyrrolidine-3-yl)-1H-benzo[d]imidazole-4-carboxamide: A potent Poly (ADP-ribose) Polymerase Inhibitor for the Treatment of Cancer” is a poorly written manuscript that is in need of major revisions prior to publication. I would recommend the following changes occur prior to publication:
1) There are significant English/grammatical errors throughout the text, please correct
2) The whole premise of the paper is that the novel PARP inhibitors described by the authors can prevent the growth of two cell lines MDA-MB-436 and CAPAN-1 in which known PARP inhibitors don’t affect? Why did the authors choose these cell lines? The authors did not provide an adequate reason. If their compounds work and known PARP inhibitors don’t, does this mean that their title compounds are operating through some other mechanism? If so, what is it?
3) The synthetic scheme, the numbering in particular, is very difficult to understand. Please use one number per compound. For example, the protected pyrrolidine ‘B1-6’ is shown as one compound, but does B1-6 actually mean 6 different compounds? If so, what are they? The experimental section seems to have multiple steps to get to N1, but the scheme only shows 2 steps to get to that point??
4) The binding studies do not show anything novel. Their compounds bind the active site in a manner similar to many other known PARP1 inhibitors. The authors did not provide an adequate explanation for why their binding mode is different/novel. I’m not sure what a “hollow existing in the binding” means. If the binding is improved in some way, why isn’t the activity appreciably different versus e.g. veliparib? Please address these concerns in the revised manuscript.
5) Is ‘membrane osmosis’ really just another term for membrane permeability? If so, please use membrane permeability, I think this is more recognizable and more used.
6) The experimental section is lacking in all of the following ways: a. What is happening in steps 1-6?? The scheme only shows one step to get to N1, not 6? b. What compound is B1-1, B1-2, B1-3, B1-4, B1-5, C1 etc. There are waaaaaaaay to many compounds listed in this experimental without the requisite structures in the text. c. There is no spectral data for any of these compounds either, please provide in the revised manuscript. d. The title of compound 5ce and 5ch is incorrect? e. Compound 5ce, 5ch, 5cp and 5cf are missing protons in the HNMR. f. Compounds 5ci and 5cj have protons listed as 0H?? All of these mistakes must be rectified in the revised version.
Author Response
Point 1: There are significant English/ grammatical errors throughout the text, please correct.
Response 1: The manuscript has been rechecked and the errors have been corrected.
Point 2: The whole premise of the paper is that the novel PARP inhibitors described by the authors can prevent the growth of two cell lines MDA-MB-436 and CAPAN-1 in which known PARP inhibitors don’t affect? Why did the authors choose these cell lines? The authors did not provide an adequate reason. If their compounds work and known PARP inhibitors don’t, does this mean that their title compounds are operating through some other mechanism? If so, what is it?
Response 2: PARP inhibitors can lead to synthetic lethal effect in BRCA-mutated cancer cells, so we chose MDA-MB-436, a BRCA-1-mutated breast cancer cell line, and CAPAN-1, a BRCA-2-mutated pancreatic cancer cell line, to measure the cell potency. Veliparib exhibits low cell potency in all kinds of cells, that is why we want to modify the structure of Veliparib in order to increase the activity. According to the docking study, the interactions of benzimidazole scaffold in the compound is similar to that of Veliparib, so the PARP inhibition potency did not increased greatly, while the cellular potency increased a lot.
Point 3: The synthetic scheme, the numbering in particular, is very difficult to understand. Please use one number per compound. For example, the protected pyrrolidine ‘B1-6’ is shown as one compound, but does B1-6 actually mean 6 different compounds? If so, what are they? The experimental section seems to have multiple steps to get to N1, but the scheme only shows 2 steps to get to that point??
Response 3: B1-6 only represents one compound. These compounds have been renamed in case of misunderstanding.
Point 4: The binding studies do not show anything novel. Their compounds bind the active site in a manner similar to many other known PARP1 inhibitors. The authors did not provide an adequate explanation for why their binding mode is different/novel. I’m not sure what a “hollow existing in the binding” means. If the binding is improved in some way, why isn’t the activity appreciably different versus e.g. veliparib? Please address these concerns in the revised manuscript.
Response 4: The sentence has been rephrased. The PARP inhibition assay indicated that the compound exhibited similar PARP inhibition activity compared with Veliparib, but the cell potency increased a lot. The most probable reason may be that the side chain increased the membrane permeability.
Point 5: Is ‘membrane osmosis’ really just another term for membrane permeability? If so, please use membrane permeability, I think this is more recognizable and more used.
Response 5: This mistake has been corrected.
Point 6: The experimental section is lacking in all of the following ways: a. What is happening in steps 1-6?? The scheme only shows one step to get to N1, not 6? b. What compound is B1-1, B1-2, B1-3, B1-4, B1-5, C1 etc. There are waaaaaaaay to many compounds listed in this experimental without the requisite structures in the text. c. There is no spectral data for any of these compounds either, please provide in the revised manuscript. d. The title of compound 5ce and 5ch is incorrect? e. Compound 5ce, 5ch, 5cp and 5cf are missing protons in the HNMR. f. Compounds 5ci and 5cj have protons listed as 0H?? All of these mistakes must be rectified in the revised version.
Response 6: The synthetic scheme has been revised. The synthetic route is according to known procedures and the reference has been added. The NMR of the compounds has been checked and mistakes have been rectified.

Reviewer 2 Report
The authors present the synthesis of a series of benzimidazole carboxamide derivatives, which are characterized by 1H-NMR, 13C-NMR and HRMS. Several compounds showed good PARP-1 and PARP-2 potency and cellular potency. Some of them displayed PARP-1 and PARP-2 inhibition potency as much as Veliparib, but much better cellular potency than that of two known PARP inhibitors, Olaparib and Veliparib, in two cell lines of MDA-MB-436 and CAPAN-1.
The manuscript is written not bad. The manuscript title, abstract, scheme, tables and figures are adequate to the content. However some details are missing:
The experimental part gives enough details about the synthetic procedures, but nothing about how PARP inhibition assay, Cell proliferation assay and molecular docking were carried out.
The synthesis is according to the known procedures, but references are missing.
There isn’t assignment of signals in the description of 1H and 13C NMR spectra. Please use bs (broad singlet) in case of exchangeable signal (NH, OH).
Most probably there are strong signals of acetone in 1H and 13C NMR spectra of 5cb and 5cc.
Additional remarks:
In the abstract few lines from the template are remained.
Scheme 1: CDI is not commonly used abbreviation. The reader should refer to the experimental section in order to understand the meaning of this abbreviation.
Author Response
Point 1: The experimental part gives enough details about the synthetic procedures, but nothing about how PARP inhibition assay, Cell proliferation assay and molecular docking were carried out.
Response 1: The procedures of PARP inhibition assay, cell proliferation assay and molecular docking in the experimental section have been added.
Point 2: The synthesis is according to the known procedures, but references are missing.
Response 2: The reference has been provided.
Point 3: There isn’t assignment of signals in the description of 1H and 13C NMR spectra. Please use bs (broad singlet) in case of exchangeable signal (NH, OH).
Response 3: The NMR of the compounds has been checked and mistakes have been revised.
Point 4: Most probably there are strong signals of acetone in 1H and 13C NMR spectra of 5cb and 5cc.
Response 4: There does exist acetone signal in 1H and 13C NMR spectra of 5cb and 5cc, and this condition has been pointed out.
Point 5: In the abstract few lines from the template are remained.
Response 5: The lines from the template have been deleted.
Point 6: Scheme 1: CDI is not commonly used abbreviation. The reader should refer to the experimental section in order to understand the meaning of this abbreviation.
Response 6: The synthetic scheme has been revised.
Reviewer 3 Report
The paper discribes the synthesis of of 2-(1-(3-(4-Chloroxyphenyl)-3-oxopropyl)pyrrolidine-3-yl)-1H-benzo[d]imidazole-4-carboxamides and their evaluation as Poly(ADP-ribose) Polymerase (PARP) inhibitors for treatment of cancer.
The work is well performed and described. The results are important to the scientists in the field of medicinal chemistry.
I suggest that the paper is accepted after some modifications are made.
Required changes/questions:
My main concerns are regarding the experimental.
-In order to fully characterise new chemical entities a full data set of 1H and 13C-NMR, mass spectrometry, IR, mp and elemental analysis is required. In the manuscript IR, mp and elemental analyses are missing. The authors should provide this data. HRMS by itself is not enough to provide an indication of the purity of the compounds, elemental analysis is a better indication of purity. Moreover, whenever column chromatography is used I would like to see also the Rf values of the products.
-The authors provide the 13C-NMR data in the supporting information and not in the experimental part of the manuscript. Please include the data also in the experimental.
-Compound N3 has a yield of 103.6%. This is irrational.
-From a reaxys search I see that compounds N1, N2 and N3 are all new to the literature. Therefore a full characterisation of these compounds is required. For known compounds a mp and 1H-NMR should be provided with a citation of the literature where they are described.
Author Response
Point 1: In order to fully characterise new chemical entities a full data set of 1H and 13C-NMR, mass spectrometry, IR, mp and elemental analysis is required. In the manuscript IR, mp and elemental analyses are missing. The authors should provide this data. HRMS by itself is not enough to provide an indication of the purity of the compounds, elemental analysis is a better indication of purity. Moreover, whenever column chromatography is used I would like to see also the Rf values of the products.
Response 1: Because of the limited funding of this project, we don’t have enough money to test the elemental analysis. In addition, the Rf value of the compounds are added into the manuscript.
Point 2: The authors provide the 13C-NMR data in the supporting information and not in the experimental part of the manuscript. Please include the data also in the experimental.
Response 2: The 13C-NMR data of the compounds has been added into the experimental section.
Point 3: Compound N3 has a yield of 103.6%. This is irrational.
Response 3: This is an error and has been revised.
Point 4: From a reaxys search I see that compounds N1, N2 and N3 are all new to the literature. Therefore, a full characterization of these compounds is required. For known compounds a mp and 1H-NMR should be provided with a citation of the literature where they are described.
Response 4: Our institute has been focusing on the discovery of novel drug with efficiency and speed, so we only care about whether the final compound is right. As long as the intermediates are confirmed by NMR, we will proceed the next synthesis immediately.
The reference literature has been provided.

Reviewer 4 Report
The manuscript describes the synthesis, characterization and assay of a series of benzimidazole carboxamides, as potential treatments of cancer. In summary, the manuscript describes the development of PARP inhibitors via the synthesis of a series of benzimidazole carboxamides and testing against a validated anticancer target. The manuscript claims good PARP-1, PARP-2 and cellular potency. The manuscript also contains a comparison of the newly synthesized compounds against two known PARP inhibitors in MDA-MB-436 and CAPAN-1 cancer cell lines. Overall, I found the manuscript to be reasonably well-written and referenced, however, some improvement in grammar is required, as well as, improvements in the quality and depth of their conclusions and methods is warranted. The following revisions are suggested.
Major Revisions:
1) The largest problem with the manuscript is the complete lack of methods, procedures and supplementary data for the PARP-1, PARP-2 and cellular potency assays. The hallmark of any quality manuscript is the appropriate description of experimental results and details that would allow other researchers to reproduce the work, and instils confidence in the rigor of the results. Unfortunately, the manuscript does not contain any description on the procedures used to perform the PARP-1/2, or cellular potency assays. Given the difficulty normally experienced when repeating PARP assays found in the literature, an extensive description of the assay methods is warranted. Additionally, I would prefer to see the raw data, and ‘assay curves’ that were used to calculate IC50’s in the supplementary data and suggest that the authors provide a full account of the experimental results for the assays.
2) For the compounds listed in Table 1, an ‘exact’ structure search using SciFinder was carried out on representative examples from Table 1, and the compounds appear to be novel. However a substructure search of similar compounds, using the same core, but R = alkyl chain returned approximately 345 substances. Some of these compounds were found to be described in the patent literature, by the same group (Yuyang Jiang; CN 108997320 A 20181214) and another 27 publications. As a result, the compounds listed in this manuscript are not entirely new, as this class of compounds is largely based on a small modification of linkers to niraparib’s benzimidazole carboxamide core, with similar substructures previously described as PARP inhibitors by other publications. However, the compounds listed appear to be novel following ‘exact’ searches.
3) Line 91 to 95, sentence starting and ending, “Compounds 5cc ….. the worst.” Comparing compounds as listed, the style/position of carbonyl group, as well as its interaction with residues in the binding pocket may effect inhibition. The influence from the length of carbon chain is not adequately persuasive. Additional docking simulation of these compounds is necessary to prove that the carbonyl group has no interaction with residues.
4) Line 95 to 96, sentence starting and ending, “5cc and 5cp ….. potency.” The conclusion seems to be correct. However, an assumption or suggestion of the reason why is preferred to illustrate what makes the -OH group decrease the potency. Additional docking simulation of these compounds may be necessary.
5) Line 97 to 99, sentence starting and ending, “Compounds with ….. activity.” This conclusion is not persuasive to me. Amines in different positions and with different substituent can’t be compared directly.
6) Line 101. Comparing 5ci, 5cj, 5cp can only indicate that substituent in the para position may not make a difference, but you’ve only made two. However, ortho substituents may reduce the number of rotatable bonds and effect potency.
7) For lines 87 to 103, if docking simulation shows no interaction between -NH- and residues, I will consider the function of a -NH- to be the same as -CH2-, and draw more conclusion from this aspect.
8) Line 109 to 111, sentence starting and ending “while nitrogen atoms ….. 5co.” The influence of -NH- can only be indicated by comparing 5cc and 5co, for 5ca and 5cb, a substituent with -C=0-CH2-Ph should be used for comparison.
9) For Table 1, I do not believe that the appropriate number of significant figures are used in reporting the percent inhibition. i.e. shouldn’t 17.61 really be reported as 18. I do not believe that the accuracy of the number can be reported to two decimal places.
10) No methods or descriptions are given for the Molecular Docking results.
11) Overall, the experimental, with respect to NMR and HRMS seems reasonably appropriate with some improvements needed, with some spectra’s having some unknown peaks in the 1.2-1.3 range. In addition, authors should address
a. I don’t really understand the B1-6 synthesis to B1 (scheme 1) section. i.e. Synthesis of N3, section 4.2.1. You have B1-1 to B1-2, to B1-3, etc. I don’t really understand this ‘scheme’. Also, what is C1, could not find this. You have no NMR’s for these compounds in this section, except B1-4 and N3. In step 6 you have, “A solution of B1…” to make B1? Step 9 is in 103.6% yield. Fixing this section (4.2.1) would improve manuscript.
12) More thorough Conclusion section? This part seems a little thin.
13) In Table 1, why weren’t all of the compound evaluated for PARP-1/2 IC50’s?
Minor Revisions:
1) Line 39. Re-word sentence, problem with the use of ‘kinds’ of PARP inhibitors. Are these inhibitors really different ‘kinds’ of PARP inhibitors? Elaborate more on what you mean by ‘kinds’. Also, the authors forgot talazoparib, recently approved PARP inhibitor.
2) Line 63. Should be ‘in the catalytic site.
3) Line 83, scheme 1 reagents, should be 2,3-diaminobenzamide-2HCl, missing letter n.
4) Line 101. 5ci, 5cj, 5cp should be in bold.
5) Line 102, IC50 should be subscript 50
6) Line 116, 5ci should be bold.
7) No units for PARP-1/2 IC50’s in Table 1
Author Response
Point 1: The largest problem with the manuscript is the complete lack of methods, procedures and supplementary data for the PARP-1, PARP-2 and cellular potency assays. The hallmark of any quality manuscript is the appropriate description of experimental results and details that would allow other researchers to reproduce the work, and instils confidence in the rigor of the results. Unfortunately, the manuscript does not contain any description on the procedures used to perform the PARP-1/2, or cellular potency assays. Given the difficulty normally experienced when repeating PARP assays found in the literature, an extensive description of the assay methods is warranted. Additionally, I would prefer to see the raw data, and assay curves that were used to calculate IC50s in the supplementary data and suggest that the authors provide a full account of the experimental results for the assays.
Response 1: The procedures of PARP inhibition assay and cell proliferation assay have been added into the manuscript. The raw data of cellular potency and preliminary screening of PARP inhibition at 10 nM is provided. The IC50 of PARP inhibition was requested, but not provided by the epiboly company.
Point 2: For the compounds listed in Table 1, an 'exact' structure search using SciFinder was carried out on representative examples from Table 1, and the compounds appear to be novel. However, a substructure search of similar compounds, using the same core, but R = alkyl chain returned approximately 345 substances. Some of these compounds were found to be described in the patent literature, by the same group (Yuyang Jiang; CN 108997320 A 20181214) and another 27 publications. As a result, the compounds listed in this manuscript are not entirely new, as this class of compounds is largely based on a small modification of linkers to niraparib's benzimidazole carboxamide core, with similar substructures previously described as PARP inhibitors by other publications. However, the compounds listed appear to be novel following 'exact' searches.
Response 2: The published patent is also based on this scaffold, but the compounds synthesized in this article were developed on the basis of the previous work.
Point 3: Line 91 to 95, sentence starting and ending, "Compounds 5cc ….. the worst." Comparing compounds as listed, the style/position of carbonyl group, as well as its interaction with residues in the binding pocket may effect inhibition. The influence from the length of carbon chain is not adequately persuasive. Additional docking simulation of these compounds is necessary to prove that the carbonyl group has no interaction with residues.
Response 3: According to the molecular docking study, the side chain of 5cd and 5cp has no interaction with residues of PARP-1, but the carbonyl group in 5cd forms a hydrogen bond with Tyr-896. However, it is not a functional residue. This hydrogen bond may not contribute to its inhibition potency, but the physical and chemical properties. The docking diagram has been provided.
Point 4: Line 95 to 96, sentence starting and ending, "5cc and 5cp ….. potency." The conclusion seems to be correct. However, an assumption or suggestion of the reason why is preferred to illustrate what makes the -OH group decrease the potency. Additional docking simulation of these compounds may be necessary.
Response 4: According to additional docking study of 5cg, -OH group forms hydrogen bond with Gly-894, while there is no hydrogen bond between imidazole nitrogen and Glu-988. Lacking this essential hydrogen bond may be the reason why the potency of the compound decreased. Since Gly-894 is not a functional residue. This hydrogen bond may not contribute to its inhibition potency, but the physical and chemical properties. The docking diagram has been provided.
Point 5: Line 97 to 99, sentence starting and ending, "Compounds with ….. activity." This conclusion is not persuasive to me. Amines in different positions and with different substituent can't be compared directly.
Response 5: The sentence has been rephrased.
Point 6: Line 101. Comparing 5ci, 5cj, 5cp can only indicate that substituent in the para position may not make a difference, but you've only made two. However, ortho substituents may reduce the number of rotatable bonds and effect potency.
Response 6: The mistake has been revised.
Point 7: For lines 87 to 103, if docking simulation shows no interaction between -NH- and residues, I will consider the function of a -NH- to be the same as -CH2-, and draw more conclusion from this aspect.
Response 7: Docking study showed that –NH- group did not have interactions with residues. And the conclusion has been revised.
Point 8: Line 109 to 111, sentence starting and ending "while nitrogen atoms ….. 5co." The influence of -NH- can only be indicated by comparing 5cc and 5co, for 5ca and 5cb, a substituent with -C=0-CH2-Ph should be used for comparison.
Response 8: The previous conclusion is not completely correct. The sentence has been rephrased to avoid the mistake.
Point 9: For Table 1, I do not believe that the appropriate number of significant figures are used in reporting the percent inhibition. i.e. shouldn't 17.61 really be reported as 18. I do not believe that the accuracy of the number can be reported to two decimal places.
Response 9: The significant figures have been revised, to one decimal place.
Point 10: No methods or descriptions are given for the Molecular Docking results.
Response 10: The procedures of molecular docking have been added into the manuscript.
Point 11: 11) Overall, the experimental, with respect to NMR and HRMS seems reasonably appropriate with some improvements needed, with some spectra's having some unknown peaks in the 1.2-1.3 range. In addition, authors should address:
a. I don't really understand the B1-6 synthesis to B1 (scheme 1) section. i.e. Synthesis of N3, section 4.2.1. You have B1-1 to B1-2, to B1-3, etc. I don't really understand this 'scheme'. Also, what is C1, could not find this. You have no NMR's for these compounds in this section, except B1-4 and N3. In step 6 you have, "A solution of B1…" to make B1? Step 9 is in 103.6% yield. Fixing this section (4.2.1) would improve manuscript.
Response 11: Additional NMR has been measured, the unknown peaks are come from the silica gel used in the chromatography, which has been provided in the supplemental material.
The experimental section has been revised. The mistakes have been corrected.
The synthetic route is according to known procedures, and the reference literature has been added.
Point 12: More thorough Conclusion section? This part seems a little thin.
Response 12: The conclusion section has been revised.
Point 13: In Table 1, why weren't all of the compound evaluated for PARP-1/2 IC50?
Response 13: The funding provided is not sufficient for the evaluation of IC50 of all the compounds, so we only evaluated some of the compounds exhibiting relatively good PARP inhibition potency after preliminary screening.

Round 2
Reviewer 1 Report
The authors made the requisite major changes that i suggested, so the only changes that need to be made are grammatical. Here are a few that I caught:
Page 1, line 35 - niacin should read nicotinamide
Page1, line 37 - inhibition of PARP enzyme should just read 'inhibition of PARP' since we all know that PARP is an enzyme
Page 1, line 40, phenantheridione should be phenanthridone?
Page 2, lines 41 and 42 - 'there are four kinds of PARP inhibitors' should read 'there are four approved PARP inhibitors'
Page 2, line 68 - I still don't know what you mean by 'hollow lacking interactions' please rephrase
Page 8, line 148 - biding should be binding
Author Response
Point 1: The authors made the requisite major changes that i suggested, so the only changes that need to be made are grammatical. Here are a few that I caught:
Page 1, line 35 - niacin should read nicotinamide
Page1, line 37 - inhibition of PARP enzyme should just read 'inhibition of PARP' since we all know that PARP is an enzyme
Page 1, line 40, phenantheridione should be phenanthridone?
Page 2, lines 41 and 42 - 'there are four kinds of PARP inhibitors' should read 'there are four approved PARP inhibitors'
Page 2, line 68 - I still don't know what you mean by 'hollow lacking interactions' please rephrase
Page 8, line 148 - biding should be binding
Response 1: The manuscript has been rechecked and the mistakes have been corrected. The sentence in line 68 has been rephrased.
Reviewer 2 Report
Prom my point of view it seems now OK!
Author Response
Point 1: Prom my point of view it seems now OK!
Response 1: Thank you for your reviewing!
Reviewer 3 Report
The paper can now be accepted for publication.
Author Response
Point 1: The paper can now be accepted for publication.
Response 1: Thank you for your reviewing!
Reviewer 4 Report
I am still not satisfied with the experimental detail for the PARP assay. It appears that the authors are performing a colorimetric PARP assay, similar to those commercially available, such as Trevigen's PARP Universal Colorimetric Assay Kit, Cat. Number 4677-096-K. Unfortunately there is insufficient detail to replicate the assay, and based on this, I would recommend further revisions.
In my opinion, the PARP assay experimental should be written in such a way that a skilled technician could replicate the assay based solely on the experimental details described in the manuscript. Including amounts of reagents, wavelengths, order of addition, concentrations, buffer, etc. (in short, a protocol that an undergrad could perform based solely on the experimental detail)
As the PARP assay is currently written, the assay could not be performed in this way and give the results stated. i.e. the plate reader has nothing to measure, the only reagents in the wells are the enzyme, DNA, inhibitor and Strep-HRP.... How are you measuring PARP activity?
The authors also state that, " The raw data of cellular potency and preliminary screening of PARP inhibition at 10 nM is provided. The IC50 of PARP inhibition was requested, but not provided by the epiboly company "
I could not find the raw data attached to this manuscript. I see the NMR and LCMS, and NMR-mnova files, but no raw data.
The manuscript is still missing the experimental for the PARP IC50's, which judging by the authors response, was performed by a company called Epiboly? There is no reference in the manuscript to this, or to a protocol. I did a search in the manuscript for 'epiboly' and found nothing.
Also, in Scheme 1 (c), it should be methyl acrylate, not acrilate
Author Response
Point 1: I am still not satisfied with the experimental detail for the PARP assay. It appears that the authors are performing a colorimetric PARP assay, similar to those commercially available, such as Trevigen's PARP Universal Colorimetric Assay Kit, Cat. Number 4677-096-K. Unfortunately there is insufficient detail to replicate the assay, and based on this, I would recommend further revisions.
In my opinion, the PARP assay experimental should be written in such a way that a skilled technician could replicate the assay based solely on the experimental details described in the manuscript. Including amounts of reagents, wavelengths, order of addition, concentrations, buffer, etc. (in short, a protocol that an undergrad could perform based solely on the experimental detail)
As the PARP assay is currently written, the assay could not be performed in this way and give the results stated. i.e. the plate reader has nothing to measure, the only reagents in the wells are the enzyme, DNA, inhibitor and Strep-HRP.... How are you measuring PARP activity?
Response 1: The experimental section of PARP inhibition assay has been revised. The experiments for the PARP inhibition at 10 nM and IC50 were performed by a CRO company called Medicilon using PARP-1 Chemiluminescent Assay Kit (BPS Bioscience, catalog 80569) and PARP-2 Chemiluminescent Assay Kit (BPS Bioscience, catalog 80552).
Point 2: The authors also state that, " The raw data of cellular potency and preliminary screening of PARP inhibition at 10 nM is provided. The IC50 of PARP inhibition was requested, but not provided by the epiboly company "
I could not find the raw data attached to this manuscript. I see the NMR and LCMS, and NMR-mnova files, but no raw data.
The manuscript is still missing the experimental for the PARP IC50's, which judging by the authors response, was performed by a company called Epiboly? There is no reference in the manuscript to this, or to a protocol. I did a search in the manuscript for 'epiboly' and found nothing.
Response 2: The raw data of cellular potency and preliminary screening of PARP inhibition at 10 nM is resubmitted. The statement " The raw data of cellular potency and preliminary screening of PARP inhibition at 10 nM is provided. The IC50 of PARP inhibition was requested, but not provided by the epiboly company " should be corrected to " The raw data of cellular potency and preliminary screening of PARP inhibition at 10 nM is provided. The IC50 of PARP inhibition was requested, but not provided by the CRO company called Medicilon"
Point 3: Also, in Scheme 1 (c), it should be methyl acrylate, not acrylate
Response 3: This mistake has been corrected.